# Racial and Ethnic Differences in Eating Duration and Meal Timing: Findings from NHANES 2011–2018

**DOI:** 10.3390/nu14122428

**Published:** 2022-06-11

**Authors:** Velarie Y. Ansu Baidoo, Phyllis C. Zee, Kristen L. Knutson

**Affiliations:** Center for Circadian and Sleep Medicine, Department of Neurology, Northwestern University Feinberg School of Medicine, Chicago, IL 60611, USA; velarie.ansu@northwestern.edu (V.Y.A.B.); p-zee@northwestern.edu (P.C.Z.)

**Keywords:** meal timing, eating duration, time-restricted eating, dietary chronotype, intermittent fasting, circadian rhythms, circadian misalignment, NHANES, health disparities

## Abstract

Background: In addition to quantity and quality, meal timing and eating duration are additional dietary characteristics that impact cardiometabolic health. Given that cardiometabolic health disparities exist among racial and ethnic groups, we examined whether meal timing and eating duration are additional diet-related differences among racial and ethnic groups. Methods: Participants (*n* = 13,084) were adults (≥20 years) from the National Health and Nutrition Examination (NHANES, 2011–2018) Survey. Times of first and last meal and the interval between them (eating duration) were derived from two 24-h dietary recalls. Multiple linear regression analyses compared these variables among race and ethnicity after adjusting for potential confounders. Results: Compared to non-Hispanic White adults, the first mealtime was significantly later for Mexican American (23 min), Non-Hispanic Asian (15 min), Non-Hispanic Black (46 min), and Other Hispanic (20 min) and Other Racial (14 min) adults (all *p* < 0.05). Mexican American and Non-Hispanic Asian adults had a significantly different last mealtime by 13 min earlier and 25 min later, respectively, compared to Non-Hispanic White adults. Compared to Non-Hispanic White adults, the mean eating duration was shorter for other Hispanic (20 min), Mexican American (36 min), and Non-Hispanic Black (49 min) adults. Conclusions: Meal timing and eating duration are additional dietary characteristics that vary significantly among racial and ethnic groups.

## 1. Introduction

In the United States, cardiovascular disease (CVD) is the leading cause of death with one person dying every 36 s [1,2]. Overweight and obesity are major risk factors for CVD. In US adults, the prevalence of obesity was 42% in 2017–2018 with Non-Hispanic Black (49.6%), Hispanic (44.8%), and Non-Hispanic White (42.2%) adults having the highest rates [3]. Overweight and obesity develop due to medical conditions and lifestyle behaviors, including diet and nutrition. Optimal nutrition plays an important role in the prevention of deaths resulting from cardiometabolic diseases [4]. According to a 2017 global study, poor dietary intake was responsible for 11 million adult deaths, which were mostly due to CVD [5]. While there is a major focus on the nutritional value of one’s diet, recent research [6,7] has emphasized the need to focus on the significance of meal timing on overall health.

One novel dietary pattern that has garnered attention is time-restricted eating (TRE). In TRE, energy intake is restricted to an eating window of between 4–10 h in a day without caloric restriction [8]. Results from clinical trials have suggested that TRE reduces risk factors for metabolic diseases [9,10,11]. A recent systematic review examined how TRE affected humans and concluded that TRE resulted in a 3% average weight loss, decreased waist circumference, reduced systolic blood pressure, and decreased cholesterol levels [12]. In addition, the timing of energy intake may also impact cardiometabolic health and evidence suggests that eating earlier in the day is associated with better metabolic outcomes as compared to later intake [7,13,14]. Further, research [15,16] has found breakfast eaters are less likely to have risk factors associated with CVD while breakfast skippers [17,18,19] have higher body mass index (BMI), which also suggests that meal timing has important cardiometabolic implications. The relationship between the timing of dietary intake and eating is known as chrono-nutrition (clock timing of nutrition). Chrono-nutrition is an emerging field that uses timed eating to improve health outcomes [6,20,21].

Cardiometabolic diseases are disproportionately experienced by racial and ethnic minority groups, and therefore, examining dietary patterns among racial and ethnic minority groups may inform our understanding of these cardiometabolic health disparities. Socioeconomic and demographic factors contribute to diet-related disparities [22], including neighborhood-built environment, economic factors, food accessibility, and availability [23,24,25,26]. Racial minority groups at risk of diet-related disparities are also more prone to diet-related chronic diseases such as CVD, obesity, and type 2 diabetes [27,28]. Here we examine whether meal timing and eating duration are additional diet-related differences among racial and ethnic groups. 

We utilized nationally representative data from the U.S. to examine the meal timing patterns among US adults by race and ethnicity. We hypothesized that eating duration and timing will vary by race and ethnicity. 

## 2. Materials and Methods

We used the cross-sectional data from the National Health and Nutrition Examination Survey (NHANES 2011–2018) for our study. The NHANES uses a complex, stratified, multistage, probability cluster sample design that is representative of the civilian non-institutionalized populations in the USA. A more detailed description of the methods used in NHANES has been described previously [29]. The demographic, physical, and anthropometric questionnaire data and the dietary recall data from four continuous 2-year cycles (i.e., 2011–2012, 2013–2014, 2015–2016, and 2017–2018) from the NHANES were used. Data for shift workers were not available for these cycles, hence we were unable to specify study participants’ work schedules.

All study participants in the NHANES provided written informed consent. The NHANES survey protocol was approved by the National Center for Health Statistics Research Ethics Review Board, Centers for Disease Control and Prevention. Our use of publicly available de-identified secondary data from the NHANES was classified as exempted by the Institutional Review Board of Northwestern University.

### 2.1. Study Measures 

The primary outcome variables were first mealtime, last mealtime, and eating duration, which is the interval between first and last meal. These outcomes are derived from dietary recalls. Participants were asked to complete 2 non-consecutive 24-h dietary recalls that included reported times of energy intake. In the NHANES data, the times for food intake are recorded as the 24 h before the interview (i.e., from midnight to midnight) as outlined in the dietary recall protocol. Since the 24-h recall captures intake from the last 24-h, there is a possibility that recall times capture energy intake from the previous night for participants who ate after midnight. For our study, we are interested in meal timing based on a single wake period, therefore, we set the start of a behavioral day to 5 am. The first and last mealtimes were recorded as meals after 5 am and before 4:59 am respectively. Thus, meals consumed between midnight to 4:59 am were considered late-night meals. The eating duration for our study is calculated as the average last mealtime minus the average first mealtime. For example, if the first meal was consumed at 7 am and the last meal was consumed at 10 pm, then the eating duration is 15 h (i.e., 22 − 7 = 15). Time was converted to hours (i.e., intake at 10:30 am was recorded as 10.5 h).

Study participants self-reported race and ethnicity, which were classified as Mexican American, Non-Hispanic Asian, Non-Hispanic Black, Non-Hispanic White, Other Hispanic, and Other Race including Biracial. Covariates that were examined in this study included age (20–39, 40–59, 60+ years), gender (female/male), marital status (married, widowed, divorced, separated, never married, living with a partner), and educational status (college and above, some college, high school, and GED, 9th to 11th grade), total energy intake (kilocalories) and body mass index (BMI). Body mass index was calculated as weight in kilograms divided by height in meters squared, and categorized as underweight (<18.5 kg/m^2^), normal (18.5–24.9 kg/m^2^), overweight (25–29.9 kg/m^2^), and obese (>30 kg/m^2^) [30].

### 2.2. Statistical Methods

We restricted our analyses to participants who were 20 years and older, had two dietary records with mealtimes, and were not pregnant. We also excluded study participants whose two dietary recalls were both on weekends because weekend days are expected to differ from weekdays but only constitute 2/7 of the week. Our final sample size was 13,084 respondents. Our primary analyses combined participants whose two dietary recalls were both weekdays with those whose dietary recalls included one weekday and one weekend. However, we also stratified our analyses between these two groups to determine whether the differences among racial and ethnic groups varied. Statistical analyses used survey analysis procedures in SAS (version 9.4; SAS Institute Inc., Cary, NC, USA). The two dietary records for each participant were averaged. Descriptive statistics for the outcome variables, first meal, last meal, and eating duration, were reported as means, standard errors (SE), standard deviations (SD), and range. Categorical variables were reported as frequencies, weighted frequencies, and weighted frequency percentages.

Multiple linear regression analyses were conducted to examine the relationship between first mealtime, last mealtime, and eating duration with race and ethnicity while accounting for the covariates. All statistical analyses accounted for the sampling weight, strata, and clusters of the NHANES data. Statistical significance was defined as a two-tailed *p*-value < 0.05 for all analyses.

## 3. Results

Descriptive statistics of the sample are summarized in Table 1. Most study participants were Non-Hispanic White (weighted proportion of 68%) and about half of the participants were female (55%). Most of the study population were married (56%), overweight and obese (71%), and had a “college and above” and some college degree (68%). Results from the NHANES 2011–2018 data showed US adults’ first and last meals were consumed at 7.9 (7:54 am) and 21.0 (9:00 pm) hours on average, respectively. Means of first mealtime, last mealtime, and eating duration by race and ethnicity are depicted in Figure 1. 

Table 2 summarizes the results from the multiple linear regression for each dietary outcome variable. Compared to non-Hispanic White adults, all racial groups had a later average first mealtime. The first mealtime was 46 min later for Non-Hispanic Black adults, 23 min later for Mexican American adults, 20 min later for Other Hispanic adults, 15 min later for Non-Hispanic Asian adults, and 14 min for other races (all *p* < 0.05). Only Mexican American and Non-Hispanic Asian adults had a significantly different last mealtime. The last meal was 13 min earlier for Mexican American adults and 25 min later for Non-Hispanic Asian adults, as compared to Non-Hispanic White adults (Table 2). Additionally, compared to Non-Hispanic White adults, other Hispanic (20 min shorter), Mexican American (36 min shorter), and Non-Hispanic Black (49 min shorter) adults had shorter eating durations (Table 2). 

There were 6139 respondents (47%) whose dietary recalls were both on weekdays and 6945 respondents (53%) who provided one weekday and one weekend dietary recall. Table 3 below presents the results from the linear regression models. Results were consistent with the findings from the two groups combined. All racial and ethnic groups had a significantly later first mealtime compared to Non-Hispanic white adults, however, the delay appeared larger when only weekdays were included in the dietary recalls. The last mealtime remained significantly later in Non-Hispanic Asian adults for both groups but was only significant in Mexican American adults whose dietary recalls included one weekend day. Finally, compared to Non-Hispanic whites, eating duration was shorter in all the racial and ethnic groups, except for Non-Hispanic Asian adults, when dietary recalls were based only on weekdays.

## 4. Discussion

Our study used nationally representative NHANES (2011–2018) data to determine whether eating duration and meal timing are additional diet-related differences among racial and ethnic groups. Results did indeed indicate racial and ethnic differences in first mealtime, last mealtime, and eating duration. Specifically, non-Hispanic White adults had an earlier first mealtime than all other racial and ethnic groups. The largest delay was among Non-Hispanic Black adults, whose average first meal was 46 min later. Mexican American adults had an earlier last mealtime while Non-Hispanic Asian adults had a later last mealtime compared to Non-Hispanic White adults. This meant that the average eating duration was shorter for non-Hispanic Black, Mexican American, and Other Hispanic adults. These differences appeared larger when dietary recalls were based only on weekdays.

These findings have implications for racial and ethnic health disparities because these dietary patterns have been linked to cardiometabolic outcomes [31,32,33]. For example, studies have found that later meal intake was associated with a higher likelihood of being overweight [34,35]. In a large weight-loss intervention study (n = 3362), late eaters had a higher BMI, triglyceride levels, and lower insulin sensitivity at baseline and lost less weight over the intervention period [33]. Meal intake in the morning has been found to be more beneficial as compared to evening intake [7,13,14,36]. For example, an observational study found that energy intake earlier in the day was significantly associated with higher insulin sensitivity [7]. Similar findings were observed in a small randomized controlled clinical trial in people with diabetes (*n* = 18); participants who consumed a high-energy breakfast had a reduction in postprandial hyperglycemia [14]. Results from a large cohort study (NHANES 1988–1994 and 1999–2014; *n* = 34,609) found early first mealtime (before ~8:00 a.m.) was associated with a small survival advantage for ≥40-year-olds and an overall reduction of metabolic risks [37]. Together, these studies indicate that eating earlier in the day has beneficial health effects.

Prior literature has also indicated that a shorter eating interval (or TRE) correlates with a lower risk of cardiometabolic diseases [9,10,11]. Eating duration for most racial groups in our study was shorter than for Non-Hispanic White adults. The shorter eating duration appears to be due to the later timing for their first meals in these groups but similar last mealtimes, which results in a shorter overall duration. 

The timing of meals both influences and is influenced by the circadian system, which regulates 24-h rhythms in myriad physiological processes, including appetite and metabolic function. The circadian system is synchronized to the 24-h day primarily by light-dark cycles that act upon the central clock in the brain, however, feeding can also synchronize internal clocks, especially ones in peripheral tissues such as the liver, gut, and muscle [38,39]. Circadian disruption can occur when there is misalignment between central and peripheral clocks or between clocks and behavior, such as feeding, and this misalignment can lead to impairment of metabolic function and risk for metabolic disease [38,39,40,41,42]. Further, late-night eating has been associated with circadian misalignment, which could lead to an increased risk of poor metabolic outcomes [43,44,45]. Some small studies have reported racial differences in the circadian system, including shorter internal clock periods and differential phase shifting in Black adults compared to White adults [46,47], however, it is unknown whether differences in underlying circadian processes among racial and ethnic groups explain differences in meal timing.

Of course, when someone eats is not entirely controlled by biology and is strongly influenced by sociocultural factors, much like other health behaviors. These socio-cultural and behavioral determinants of meal timing include breakfast skipping [7,14,48,49], type of work such as shiftwork [50,51], and socioeconomic factors [49]. Skipping breakfast is one cause of later meal timing; adults who skipped breakfast consumed a greater amount of food at lunch and dinner [48] and breakfast consumption is associated with higher socioeconomic status [49]. The benefits of consuming breakfast have been well-documented [52,53], however, many adults skip breakfast worldwide [49,54,55,56]. Non-Hispanic Whites and Non-Hispanic Blacks have been reported as the groups that are most likely and least likely to consume breakfast, respectively [49]. 

The nature of one’s job can also influence meal timing. For example, many essential workers (e.g., grocery workers, healthcare, transportation) are shift workers who have variable eating patterns, including later meal timing, due to the [57] timing of their work shift [50,51]. Racial and ethnic minority groups are overrepresented in essential jobs and are more likely to be shift workers as compared to traditional workers [50,51]

Individuals who are shift workers are at increased risk of many chronic diseases. For example, Black female nurses who are rotating night shift workers were found to have higher incident hypertension risk as compared to Black female nurses who did not work the night shift. Interestingly, there was no increase in risk for White female nurses [58]. Therefore, differences in job type and characteristics may partly explain the later average first meal timing in all the underrepresented racial and ethnic groups in our study.

The strength of our study includes the use of a large, nationally representative sample of the U.S adult population to compare meal timing in racial and ethnic groups in the U.S. Our use of a two-day dietary intake time minimizes error in estimating participants’ usual intake times compared to a single 24-h recall [59]. Our study had some limitations, which include not having information on which participants were shift workers. Additionally, we did not have measures of participants’ sleep and internal circadian timing [35,60]. Thus, we could not identify first and last mealtimes relative to their sleep or endogenous circadian periods. We hope that by resetting the 24-h dietary recall times (originally midnight to midnight) to the start of the behavioral clock times (i.e., first mealtime was defined as after 5 a.m.) we minimized errors due to identifying a late-night meal (e.g., 1:00 a.m.) as a first meal. Lastly, self-reported times of intake are prone to participants’ memory hence there is a likelihood that inaccurate times may be reported by the study participant. By estimating multiple-day times and the use of the multiple pass method in the NHANES dietary recall methods, this error may be minimized [59,61]. 

Overall, the findings of our study showed that Mexican American, Non-Hispanic Asian, Non-Hispanic Black, Other Hispanic adults, and Other racial groups had later first mealtimes as compared to Non-Hispanic White adults. Last mealtimes only differed for Mexican American and Non-Hispanic Asian adults (compared to non-Hispanic White adults). Non-Hispanic Black adults had the shortest eating duration as compared to non-Hispanic White adults. Our study highlights the potential importance of racial differences in meal timing and its possible role in disparities in cardiometabolic health. Meal timing is a potential novel modifiable risk factor for CVD that could be developed for different racial and ethnic groups. Future studies should examine the role of meal timing in racial and ethnic disparities in cardiometabolic diseases. 

## Figures and Tables

**Figure 1 nutrients-14-02428-f001:**
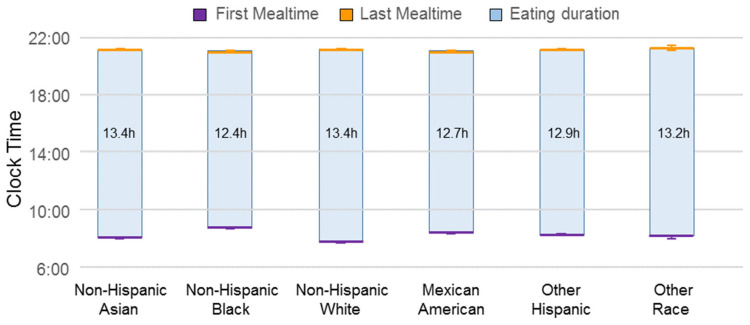
First mealtime (bottom of bar), last mealtime (top of bar), and eating duration (length of bar) by race and ethnicity.

**Table 1 nutrients-14-02428-t001:** Descriptive characteristics of the U.S (United States) adult population, NHANES 2011–2018.

Variable	*n*	Frequency	Weighted %
Gender Male Female	13,084	6289 6795	47 53
Race and Ethnicity Mexican American Non-Hispanic Asian Non-Hispanic Black Non-Hispanic White Other Hispanic Other Race	13,084	1328 1447 3127 5501 1195 486	6.3 4.4 11.1 69.7 5.2 3.3
Age 20–39 40–59 60+	13,084	4316 4460 4308	33.3 38 28.7
Education 9th–11th grade High School and GED Some College College and above	13,084	1750 3155 4408 3771	9.5 22.5 33.7 34.3
Marital Status Married Widowed Divorced Separated Never married Living with partner	13,084	6668 907 1494 407 2559 1044	56.1 5.3 10.6 2.2 18.1 7.7
BMI Underweight Normal weight Overweight Obese	13,084	208 3497 4121 5136	1.5 26.9 32.4 39.2
	*n*	Mean	Range
Mean first meal (hrs.)	13,084	7.91	5–22.50
Mean last meal (hrs.)	13,084	21.13	10.0–28.96
Mean eating duration (hrs.)	13,084	13.22	0–23.87
Mean total energy (kcal)	13,084	2083.54	96.50–10025

First meal refers to the mean first time of intake (after 5 am); last meal refers to the mean last time of intake for the day; eating duration was calculated as the first mealtime subtracted from the last mealtime. All participants self-reported two days of intake times. Behavioral time for the day starts at 5 am and ends at 4.59 am. Descriptive statistics calculations were based on weighted estimates from the NHANES: hrs. = hours. Unweighted standard deviations (SD) for first mealtime, last mealtime, and eating duration are 1.8, 2.4, and 3.1 respectively. Sample size per cycle 2011–2012 (*n* = 3766, 28%); 2013–2014 (*n* = 4041, 30%); 2015–2016 (*n* = 3608, 29%); 2017–2018(*n* = 1669, 13%).

**Table 2 nutrients-14-02428-t002:** Multiple Linear Regression Models for First Meal, Last Meal, and Eating Duration, NHANES 2011–2018.

	First Meal (h)		Last Meal (h)		Eating Duration (h)	
Variable	Estimate (SE)	95% CI	Estimate (SE)	95% CI	Estimate (SE)	95% CI
Race and Ethnicity White (Ref) Mexican American Non-Hispanic Asian Non-Hispanic Black Other Hispanic Other Race/Biracial	0.39 (0.07) 0.25 (0.07) 0.76 (0.05) 0.34 (0.07) 0.24 (0.11)	0.26–0.53 ^a^ 0.12–0.39 ^b^ 0.65–0.86 ^a^ 0.20–0.48 ^a^ 0.03–0.45 ^c^	−0.21 (0.09) 0.41(0.09) −0.05 (0.08) −0.03 (0.08) 0.04 (0.17)	-0.39–0.03 ^c^ 0.23–0.58 ^a^ −0.21–0.10 −0.16–0.15 −0.30–0.39	−0.60 (0.11) 0.15 (0.12) −0.81 (0.10) −0.34 (0.11) −0.20 (0.21)	−0.82–−0.38 ^a^ −0.08–0.39 −1.01–−0.61 ^a^ −0.56–−0.13 ^b^ −0.62–0.23
Gender Male (Ref) Female	−0.05 (0.04)	−0.13–0.04	−0.02 (0.06) ^a^	−0.15–−0.10	0.02 (0.07)	−0.12–−0.16
Age, years 20–39 (Ref) 40–59 60+	−0.55 (0.05) −0.48 (0.05)	−0.65–−0.45 ^a^ −0.59–−0.36 ^a^	0.13 (0.07) 0.07 (0.09)	−0.003–0.27 −0.11–0.25	0.68 (0.09) 0.55 (0.10)	0.50–0.86 ^a^ 0.35–0.75 ^a^
Educational status College and above (Ref) Some College High School and GED 9th–11th grade	0.02 (0.04) 0.10 (0.05) 0.30 (0.07)	−0.06–0.11 −0.04–0.21 0.17–0.44 ^a^	0.02 (0.07) −0.03 (0.09) −0.01 (0.09)	−0.12–−0.16 −0.21–0.15 −0.20–−0.17	−0.01 (0.09) −0.13 (0.11) −0.33 (0.11)	−0.18–0.16 −0.35–−0.09 −0.32–−0.11 ^b^
Marital Status Married (Ref) Widowed Divorced Separated Never married Living with a partner	−0.005 (0.06) 0.13 (0.07) 0.45 (0.13) 0.57 (0.05) 0.40 (0.06)	−0.12–0.13 −0.002–0.26 0.18–0.72 ^b^ 0.47–0.67 ^a^ 0.26–0.53 ^a^	0.09 (0.14) 0.22 (0.12) 0.20 (0.21) 0.25 (0.08) 0.34 (0.11)	−0.20–0.37 −0.01–0.45 −0.22–0.61 0.09–0.41 ^b^ 0.11–0.56 ^b^	0.08 (0.17) 0.09 (0.12) −0.26 (0.22) −0.32 (0.10) −0.06 (0.14)	−0.26–0.42 −0.15–0.33 −0.70–0.19 −0.51–−0.13 ^b^ −0.34–0.21
BMI (kg/m^2^) Normal weight (Ref) Underweight Overweight Obese	0.42 (0.16) −0.02 (0.05) 0.03 (0.05)	0.10–0.74 ^b^ −0.11–0.08 −0.07–0.13	0.24 (0.27) −0.05 (0.07) −0.03 (0.07)	−0.31–0.78 −0.18–0.09 −0.17–0.12	−0.18 (0.32) −0.03 (0.08) −0.06 (0.09)	−0.83–0.47 −0.19–0.14 −0.24–0.13
*R* ^2^	0.10		0.04		0.07	

Multiple regression weighted estimate; number of observations = 13,084; sum of weight = 145,087,614. Ref = reference group. Significant levels two-tailed: ^a^, *p* < 0.001; ^b^, *p* < 0.01; and ^c^, *p* < 0.05. All models accounted for total energy intake.

**Table 3 nutrients-14-02428-t003:** Multiple Linear Regression Models for First Meal, Last Meal, and Eating Duration, NHANES 2011–2018 stratified by dietary recall cays.

		First Meal (h)		Last Meal (h)		Eating Duration (h)	
	Variable	Estimate (SE)	95% CI	Estimate (SE)	95% CI	Estimate (SE)	95% CI
One weekend One weekday	Race and Ethnicity White (Ref) Mexican American Non-Hispanic Asian Non-Hispanic Black Other Hispanic Other Race/Biracial	0.28 (0.09) 0.22 (0.10) 0.71 (0.08) 0.27 (0.09) −0.03 (0.15)	0.10–0.45 ^b^ 0.20–0.42 ^b^ 0.54–0.88 ^a^ 0.08–0.46 ^b^ −0.34–0.28	−0.23 (0.09) 0.46 (0.12) −0.06 (0.09) 0.12 (0.11) 0.02 (0.19)	−0.43–0.03 ^c^ 0.21–0.71 ^b^ −0.25–0.12 −0.10–0.35 −0.36–0.40	−0.51 (0.14) 0.24 (0.17) −0.77 (0.13) −0.15 (0.15) 0.05 (0.29)	−0.78–−0.24 ^a^ −0.10–0.58 −1.03–−0.51 ^a^ −0.45–−0.16 −0.54–0.64
Two Weekdays	Race and Ethnicity White (Ref) Mexican American Non-Hispanic Asian Non-Hispanic Black Other Hispanic Other Race/Biracial	0.54 (0.09) 0.29 (0.07) 0.84 (0.07) 0.43 (0.10) 0.57 (0.11)	0.35–0.73 ^a^ 0.14–0.44 ^a^ 0.70–0.97 ^a^ 0.23–0.63 ^a^ 0.25–0.89 ^a^	−0.19 (0.16) 0.32 (0.14) −0.07 (0.11) −0.18 (0.13) 0.04 (0.27)	−0.51–0.13 0.04–0.60 ^b^ −0.28–0.14 −0.44–0.08 −0.50–0.58	−0.73 (0.19) 0.03 (0.13) −0.91 (0.13) −0.61 (0.15) −0.54 (0.26)	1.10–−0.35 ^a^ −0.26–0.31 −1.16–−0.65 ^a^ −0.92–−0.30 ^a^ −1.06–−0.01 ^c^

Multiple regression weighted estimate; number of observations (one weekday one weekend = 6945; sum of weight = 79,058,463); (Two weekdays *n* = 6139; sum of weights = 66,029,151). Ref = reference group. Significant levels two-tailed: ^a^, *p* < 0.001; ^b^, *p* < 0.01; and ^c^, *p* < 0.05. All models accounted for gender, age, educational status, BMI, marital status, and total energy intake.

## Data Availability

The data that support the findings of this study are available from the first author, VAB, upon reasonable request.

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
