# Peer review of "Racial and Ethnic Differences in Eating Duration and Meal Timing: Findings from NHANES 2011–2018"

_nutrients, 2022, doi:10.3390/nu14122428_

Round 1

Reviewer 1 Report

Thanks to the editor for the invitation. In this study, the authors have investigated the Racial and Ethnic Differences in Eating Duration and Meal Timing in NHANES. This is a very interesting and well-prepared manuscript. I don’t have major concern on this manuscript.

Please see my comments below:

  1. A stratified analysis by survey year may be needed, is there any other studies have reported the trends of eating duration and meal timing in NHANES?
  2. The eating duration and meal timing on workdays should be differ from weekend. A sensitivity analysis by excluding participants whose dietary recall was conducted in weekend is needed.
  3. The start of a behavioral day at 4:00 am may be too early, this may be due to night shift workers or other special reasons (i.e., insomnia).
  4. It would be interesting to perform a stratified analysis by age, the elderly individual may have an earlier breakfast than others.

Author Response

Reviewer 1:

Thanks to the editor for the invitation. In this study, the authors have investigated the Racial and Ethnic Differences in Eating Duration and Meal Timing in NHANES. This is a very interesting and well-prepared manuscript. I don’t have major concern on this manuscript. 

Comment: A stratified analysis by survey year may be needed, is there any other studies have reported the trends of eating duration and meal timing in NHANES? 

Response: We agree that checking each cycle will be important. We therefore tested for the interaction between each of the NHANES cycles with the racial and ethnic groups. We found no statistical difference between the cycles and the racial groups. The proportion by each cycle are reported beneath the table in lines 176 and 177. “Sample size per cycle 2011-2012 (n=3766, 28%); 2013-2014 (n=4041, 30%); 2015-2016 (n=3608, 29%); 2017-2018(n=1669, 13%)”.

Comment: The eating duration and meal timing on workdays should be differ from weekend. A sensitivity analysis by excluding participants whose dietary recall was conducted in weekend is needed.

Response: The reviewer raises an important point. We agree that day of recall can affect the timing of dietary intake. As such, we examined which dietary recalls were from weekends (Saturday or Sunday) and weekdays (Monday through Friday). The proportion for two weekend days, two weekdays, and one weekend/one weekday were 3%, 43%, 51% respectively. Considering these proportions, and to be able to obtain a more representative estimate, we excluded (Lines 123,129) participants whose dietary recall days were both weekends in our analyses.  In addition, we conducted additional analyses stratifying the two remain groups (2 weekdays, 1 weekday and 1 weekend) to see if racial and ethnic differences differed. We have added another Table (Table 3) and discussed these results in the manuscript.

Comment: The start of a behavioral day at 4:00 am may be too early, this may be due to night shift workers or other special reasons (i.e., insomnia). 

Response: We changed the start of the behavioral day to 5 am and reran all analyses.  The results are reported in Tables 1 and 2. We also removed the citation that supported our selection of the 4AM cutoff time (Line 103).

Comment: It would be interesting to perform a stratified analysis by age, the elderly individual may have an earlier breakfast than others. 

Response: Thank you, we acknowledge that age is a well-established predictor of circadian timing and we adjusted for age in our models. However, we do not think that age explains the differences by race and ethnicity. Additionally, we checked if age varied by race and ethnicity and found no difference.

Reviewer 2 Report

Valarie Y. Ansu Baidoo et al. have performed an interesting analysis of NHANES database investigating chrono-nutritional habits in relation to ethnic differences. As known, chono- nutrition is an innovative aspect of nutrition research and above all is based on a different new therapeutical treatment of obesity, as reported by authors in introduction section.

The analysis is well designed, and the article is well-arranged, I have some suggestions:

1. Female and age are two covariates analysed by authors. As reported in literature, menopause is associated with disturbances in circadian rhythms and consequently an increased risk of cardiometabolic diseases (PMID: 30401550, PMID: 27585541). If possible, the authors could further analyse female group differentiating postmenopausal woman.  

2. Why the authors did not analyse energy intake? This data could be interesting (as reported by the same authors in introduction section lines 48-47): meal timing-ethnic difference influences energy intake?

3. The authors should add a scheme of chrono-eating day (line 92-95)

4. In discussion section, authors reported an important limitation of this analysis: “not having information on which participants were shift workers (line 222)”, this information should be reported also in methods section.

Author Response

Reviewer 2:

Velarie Y. Ansu Baidoo et al. have performed an interesting analysis of NHANES database investigating chrono-nutritional habits in relation to ethnic differences. As known, chrono- nutrition is an innovative aspect of nutrition research and above all is based on a different new therapeutical treatment of obesity, as reported by authors in introduction section.  

The analysis is well designed, and the article is well-arranged, I have some suggestions: 

Comment:   Female and age are two covariates analysed by authors. As reported in literature, menopause is associated with disturbances in circadian rhythms and consequently an increased risk of cardiometabolic diseases (PMID: 30401550, PMID: 27585541). If possible, the authors could further analyse female group differentiating postmenopausal woman.  

Response: We agree with the reviewers’ comment on the association between postmenopausal women and circadian rhythm disturbances as well as cardiometabolic diseases. However, we do not believe that menopause would explain our findings of differences by race and ethnicity, which is the focus of this paper. Further, we included age in our models, which is, of course, highly correlated with menopausal status, and therefore adding menopausal status to the models would not improve the fit.  

Comment: Why the authors did not analyse energy intake? This data could be interesting (as reported by the same authors in introduction section lines 48-47): meal timing-ethnic difference influences energy intake?

Response: We have added total energy intake for study participants as a covariate in all of our regression models.

Comment: The authors should add a scheme of chrono-eating day (line 92-95)  

Response: Our figure presents the average eating start time, end time and duration by race and ethnicity, which is a schematic of a chrono-eating day per group. 

Comment:  In discussion section, authors reported an important limitation of this analysis: “not having information on which participants were shift workers (line 222)”, this information should be reported also in methods section.  

Response: We have added this to the Methods section. Please see Lines 86-88. “Data for shift workers were not available for these cycles, hence we were unable to specify study participants work schedules”.

Round 2

Reviewer 2 Report

I appreciated the additional work of the authors.